# Two-Step Identification of N-, S-, R- and T-Cytoplasm Types in Onion Breeding Lines Using High-Resolution Melting (HRM)-Based Markers

**DOI:** 10.3390/ijms24021605

**Published:** 2023-01-13

**Authors:** Ludmila Khrustaleva, Mais Nzeha, Aleksey Ermolaev, Ekaterina Nikitina, Valery Romanov

**Affiliations:** 1Center of Molecular Biotechnology, Russian State Agrarian University-Moscow Timiryazev Agricultural Academy, 49, Timiryazevskaya Str., 127550 Moscow, Russia; 2All-Russian Research Institute of Agricultural Biotechnology, Timiryazevskaya 42 Str., 127550 Moscow, Russia; 3Federal Scientific Vegetable Center, Selectionaya St. 14, VNIISSOK, Odintsovo Region, 143072 Moscow, Russia

**Keywords:** cytoplasmic male-sterility, high-resolution melting (HRM), molecular markers, mitochondrial genes, onion (*Allium cepa* L.)

## Abstract

High-resolution melting (HRM) analysis is a powerful detection method for fast, high-throughput post-PCR analysis. A two-step HRM marker system was developed for identification of the N-, S-, R- and T-cytoplasms of onion. In the first step for the identification of N-, S- and R-cytoplasms, one forward primer was designed to the identical sequences of both *cox1* and *orf725* genes, and two reverse primers specific to the polymorphic sequences of *cox1* and *orf725* genes were used. For the second step, breeding lines with N-cytoplasm were evaluated with primers developed from the *orfA501* sequence to distinguish between N- and T-cytoplasms. An amplicon with primers to the mitocondrial *atp9* gene was used as an internal control. The two-step HRM marker system was tested using 246 onion plants. HRM analysis showed that the most common source of CMS, often used by Russian breeders, was S-cytoplasm; the rarest type of CMS was R-cytoplasm; and the proportion of T-cytoplasm among the analyzed breeding lines was 20.5%. The identification of the cytoplasm of a single plant by phenotype takes from 4 to 8 years. The HRM-based system enables quick and easy distinguishing of the four types of onion cytoplasm.

## 1. Introduction

Onion (*Allium cepa* L.) is the second most valuable vegetable crop in the world following tomato. A significant increase in yield through heterosis was achieved by the discovery of CMS (Cytoplasmic Male Sterility) in onion [1]. The seed propagation of male sterile lines has become an important tool for the creation of commercial F${}_{1}$ hybrids. The discovery of CMS systems in onions allows breeders to avoid the laborious work of anther removal when creating new breeding lines and overcome the difficulties in producing hybrid onion seeds. Knowledge about cytoplasm types plays an important role in onion breeding.

CMS is determined by aberrant mitochondrial genes and associated with the failure to produce functional pollen. Due to advances in plant cell molecular biology, the knowledge gained over the past decade clarified the role of mitochondria in triggering death of the male reproductive organs.

Mitochondria are the energy stations of cells. Pollen development is an energy costly process, and therefore disturbances in mitochondrial functions could have dramatic effects on male fertility [2,3,4]. CMS products may interrupt either the assembly or functions of complexes in the electron transport chain [5,6] and perhaps play roles for programmed cell death and an excessive accumulation of reactive oxygen species (ROS) that are byproducts during the operation of the electron transport chain in mitochondria [7,8,9]. It should be stressed that, due to a unique natural phenomenon of CMS, only male gametes are impaired, while female gametes are functional. Furthermore, CMS products do not show abnormal phenotypes in somatic cells, which may be due to the modification of CMS products at different stages resulting in the neutralization of their effect on mitochondrial function [10].

In plants, each mitochondrion contains multiple DNA molecules of different sizes that are capable of recombination [11]. The plant mitochondrial genome (mtDNA) is much larger than that of other eukaryotes and evolves rapidly in structure [12]. Moreover, the plant mtDNA is composed of a mixture of circular subgenomes and linear and branched molecules. Recent NGS sequencing and new technology of mitochondrial genome assembly revealed primarily non-circular forms, which could be intermediates in replication or recombination [13]. The CMS-associated genes cloned thus far have been created mostly using repeat-mediated mtDNA rearrangements. The plant mtDNA is rich in repeated sequences, which can be involved in homologous recombination events and consequently have a major impact on the structure of the mtDNA [12].

CMS-associated genes are often chimeric genes composed of partial sequences of known mitochondrial genes and unknown sequences [4,14,15]. In most cases, multiple recombination events involving known mitochondrial genes as well as sequences of unknown origin create new open reading frames (ORF) associated with cytoplasmic male sterility in higher plants [4,16]. These CMS-associated ORFs have been used for producing molecular markers to identify the type of CMS.

The first source of onion CMS was discovered by Henry A. Jones, a geneticist who was known as the ”father of the hybrid onion”, in the cultivar ’Italian Red’ in 1925, and this was later determined as a S-cytoplasm CMS [1]. The second source of CMS (T-cytoplasm) was discovered by Berninger [17] in the cultivar ’Jaune paille des Vertus’. There are two hypotheses for the origin of S-cytoplasm. According to one hypothesis, mitochondria from another Allium species entered the cytoplasm of onion via the triploid top-setting onion ’Pran’ [18]. Another hypothesis is based on the assumption of the existence of sympatric S- or N-cytoplasmic variants containing the corresponding mitochondrial DNA in the ancestral forms of the onion. S- and N-cytoplasm could be preserved during the domestication and dispersal of onion from its center of origin in Central Asia throughout the world [19].

S-cytoplasm is widely used by seed growers due to stable male sterility, no reduction in female fertility and the relatively frequent occurrence of a recessive allele (*ms*) at the nuclear locus for restoring male fertility (*Ms*), thereby, allowing seed propagation of male-sterile lines [20]. T-cytoplasm is used less commonly than S-cytoplasm [21], and presumably it is a relatively recent cytoplasmic variant closely related to the N-cytoplasm of onion [22].

Onion is a biennial crop, and the identification of cytoplasm types takes 4–8 years. However, the use of molecular markers is capable of speeding up the process and saving time for breeders. Molecular markers to distinguish mitotypes in onion have been developed [23,24,25,26,27]. Most of the markers clearly distinguish normal N-cytoplasm and sterile S-cytoplasm.

Sato [24] developed a marker that was based on the *cob* mitochondrial gene and chimeric gene and possessed an insertion of a chloroplast DNA sequence into the upstream region of *cob* in the S-cytoplasm. The Sato [24] marker distinguishes between N- and S-cytoplasm. However, the 5’ *cob*-marker does not distinguish T-cytoplasm from N-cytoplasm, and, in both cytoplasm types, the 180-bp fragment is amplified. For the first time, a marker capable of identifying N-, S- and T-cytoplasm was reported by Engelke et al. [25]. The authors proposed the combination of the 5’ *cob*-marker with the *orfA501* marker, which was developed from a CMS${}_{1}$-specific sequence in chives [28].

Later, Kim et al. [26] developed markers that also make it possible to identify three types of cytoplasm based on DNA sequences of other mitochondrial genes, the *cox1* gene encoding cytochrome c oxidase subunit I and its chimeric *orf725* gene. Recently, Havey and Kim published a comprehensive analysis of commercially used sources of CMS and the nuclear male-fertility restoration *Ms* locus [19]. The authors made an important conclusion that there was a fourth type of cytoplasm, which they proposed to be labeled as ’R’ cytoplasm, because it may have originated from the onion cultivar ’Rijnsburger’ in the Netherlands. *Bona fide* T-cytoplasm was proposed to be labeled as ’T’, which was genetically characterized by Berninger [17].

To add to this study and fill the gap covering the territory of the Russian Federation, we analyzed the types of CMS and genotyped for the nuclear *Ms* locus for fertility restoration across the onion genetic collection kindly provided by the Federal Scientific Vegetable Center and Timofeev breeding station of Russian State Agrarian University. In order to facilitate the monitoring of cytoplasmic type in large populations of breeding lines, HRM markers have been developed that clearly distinguish between N-, T-, R- and S-cytoplasms. Our study showed that S-cytoplasm is mainly used for onion breeding in the Russian Federation, while T-cytoplasm is used less often, and R-cytoplasm is used rarely.

## 2. Results

### 2.1. Development of HRM Markers for the Identification of N-, S- and R-Cytoplasms (First Step)

Each plant cell contains many mitochondria [29], and each mitochondrion contains numerous DNA molecules [11]. Stoichiometries of *cox1* and *orf725* genes in plant cell define N-, S- and R-cytoplasm types [19,26,30]. In N-cytoplasm, the copy number of normal *cox1* gene is high, and the copy number of *orf725* is nearly undetectable. In contrast, S-cytoplasm is characterized by a higher number of *orf725* genes and significantly reduced numbers of the normal *cox1* genes. R-cytoplasm contains similar relative numbers of both *cox1* and *orf725* genes [19].

In order to distinguish between these three types of cytoplasm, we developed markers for the *cox1* and *orf725* mitochondrial genes. To facilitate screening a large number of samples from onion breeding lines, we applied high-resolution melting (HRM) analysis, which is a powerful detection method for fast, high-throughput post-PCR analysis [31,32]. Previously, Kim and Kim [30] developed HRM markers based on *cox1* and *orf725* genes to identify the type of onion cytoplasm. However, the markers did not clearly distinguish between S- and T-type cytoplasms. According to later studies, the ’T-like’ type of cytoplasm carrying the *orf725* gene was proposed to be the R-type, and cytoplasm carrying *orfA501* was considered *bona fide* T-cytoplasm [19].

To develop HRM markers, we performed partial sequencing of regions with a length 594 bp based on known *cox1* gene sequences extracted from the complete mitochondrial genome of N-cytoplasm (GenBank: AP018390.1) and with a length 577 bp based on *orf725* gene sequences extracted from the CMS-S complete mitochondrial genome (GenBank: NC_030100.1) in eleven breeding lines (Appendix A). The *cox1* amplicons were obtained from eight breeding lines as expected. The *orf725* amplicons were obtained for six breeding lines possessing CMS. The amplicons were sequenced using the Sanger method with constructed primers (see Materials and Methods). Both *cox1* and *orf725* gene partial amplicons showed no difference from the normal and CMS-S references sequences.

An HRM marker was developed using a combination of one common primer (AcM-HRM-F1) anchored to the identical sequences of both *cox1* and *orf725* genes and two specific primers (AcM-HRM-R1 and AcM-HRM-R2) anchored to the polymorphic sequences of *cox1* and *orf725* genes, respectively, (Figure 1). The melting temperature (T${}_{m}$) and size of the expected amplicon from the *cox1* gene were 80 °C and 128 bp, respectively. The T${}_{m}$ and size of the expected amplicons from the *orf725* gene were 89 °C and 132 bp, respectively.

Normalized melting curves of amplicons belonging to normal cytoplasm were clearly separated from those of CMS cytoplasm (Figure 2). The curve with one peak at T${}_{m}$ = 80 °C corresponds to N-cytoplasm, the curve with one peak at T${}_{m}$ = 89 °C corresponds to S-cytoplasm, and the curve with two peaks corresponds to R-cytoplasm. The sequenced breeding lines were evaluated using the developed HRM marker system.

Thus, in the first step, using three HRM-primers (AcM-HRM-F, AcM-HRM-R1 and AcM-HRM-R2), we identified N-, S- and R-types of cytoplasm. However, these primers constructed on the *cox1* gene cannot distinguish between N (normal) cytoplasm and CMS-T, where sterility is caused by rearrangement in other unknown mitochondrial genes.

### 2.2. Developing HRM Markers for Distinguishing between N- and T-Cytoplasm Types (Second Steps)

The third type of CMS is T-cytoplasm, which is closely related to the N-cytoplasm of onion [19]. The first PCR markers based on mitochondrial DNA sequences [24] did not distinguish between N- and T-cytoplasms, while T-cytoplasm showed the same PCR product as N-cytoplasm. A PCR marker was developed to distinguish between N- and T-types of the cytoplasm by Engelke et al. [25]. This marker was developed from a CMS${}_{1}$-specific sequence in chives (*Allium schoenoprasum* L.)—a closely related species of onion [28].

In chives, the sequence is of chimerical nature, consisting of the *atp9* homologous sequence interrupted at position 147 bp and a 623 bp insertion of unknown origin at its 5’-end. Assuming that *A. cepa* CMS is of alloplasmic origin [18], the authors designed primers based on the chives sequence that span nearly the complete *orfA501*. With this *orfA501* marker, in combination with markers developed by Sato [24], it became possible to distinguish between N-, T- and S-cytoplasm types. Later, a new open reading frame mitochondrial chimeric gene (*orf725*) was isolated containing almost the entire *cox1* gene sequence at the 5’ end, a 473 bp sequence homologous to the chives *orfA501* gene and a unique sequence at the 3’ end [26]. The authors reported that it was impossible to distinguish between N- and T-cytoplasm with the *orf725* marker.

Therefore, considering all the knowledge described above, we may conclude that the *orf725* chimeric gene is the result of an insertion into *cox1* gene and that this chimeric gene is present in S- and R-cytoplasm. However, it is still unknown in which gene the *orfA501* was inserted in the onion mitochondrial genome.

In order to test the hypothesis that the *orfA501* is inserted into the *atp9* gene in *A. cepa*, as was found in *A. schoenoprasum* [28], we constructed a putative chimeric gene *in silico* by inserting a 473 bp *orfA501* sequence of the *A. cepa* from the CMS-S complete mitochondrial genome (GenBank: NC_030100.1) into the *atp9* gene of the *A. cepa* N-cytoplasm from the complete mitochondrial genome (GenBank: AP018390.1) after position 147 at the 5${}^{\prime }$-end. The HRM marker was developed using a combination of one common primer atp9-HRM anchored to the identical sequences of both *atp9* and a putative chimeric *orfA501* genes as well as two specific primers atp9-R1 and orfA501-R2 anchored to the polymorphic sequences. A PCR product with expected size and T${}_{m}$ was obtained only for the *atp9* gene suggesting its intact nature at least in the amplified region. Thus, HRM-markers were developed separately using two atp9-HRM primers designed on the *atp9* gene as an internal control and two T-HRM primers designed on the *orfA501* sequence (see Materials and Methods; Figure 3).

The normalized melting curve of amplicons belonging to the N-cytoplasm was clearly separated from that of the CMS-T cytoplasm (Figure 4a). The N-cytoplasm is characterized by only one normalized melting peak that is obtained on the *atp9* gene, and T-cytoplasm is characterized by two normalized melting peaks that are obtained on *orfA501* and the intact *atp9* (Figure 4b).

Taken all together, four types of onion cytoplasm can be distinguished by a two-step HRM-marker system. In the first step, the N-, S- and R-types of cytoplasm can be determined using primers on the *cox1* gene and its chimeric gene (*orf725*). In the second step, N- and T-types of cytoplasm can be separated using T-HRM primers anchored to the *orfA501* gene and atp9-HRM primers as an internal control.

### 2.3. Validation of the Reliability of the Two-Step HRM Marker System by Comparison with Existing PCR Marker Systems

In order to evaluate the HRM markers developed in this study, a conventional PCR was performed with the markers reported earlier by Sato [24], Engelke et al. [25] and Kim et al. [26]. The PCR markers used for the determination of cytoplasm types are presented in Materials and Methods section below. The Sato [24] marker distinguished N- and S-cytoplasms. The combination of the Sato [24] marker with the *orfA501* marker proposed by Engelke et al. [25] distinguished three types of cytoplasm: N-, S- and *bona fide* T-cytoplasms. The primers developed by Kim et al. [26] identified N-, S- and R-cytoplasms [19]. Using the combination of three marker systems, it is possible to distinguish four cytoplasm types of onion (Table 1). The analysis of eleven breeding lines with conventional PCR showed that the results (Table 2; Figure 5) completely agreed with the HRM results.

### 2.4. Analysis of the Cytoplasm Types and Ms Locus in Breeding Lines Used by Russian Breeders

A total of 246 individual plant DNA samples from 77 breeding lines were analyzed using the two-step HRM marker system developed in this study. For the genotyping of the *Ms* locus, we used Rf-HRM7 markers developed by Kim and Kim [30]. The authors constructed Rf-HRM7 markers based on InDel in partial sequences of *AcPMS1* alleles.

The *AcPMS1* gene encoding the PMS1 protein involved in DNA mismatch repair in other plant species was suggested to be a causal gene for the restoration of male-fertility [33]. The identification of cytoplasm type and genotyping at the *Ms* locus showed that 73 plants possessed N-cytoplasm and were homozygous recessive at the *Ms* locus (*msms*) and, therefore, are maintainer lines (Table 3). Twelve plants among 100 analyzed plants with N-cytoplasm were scored as homozygous dominant (*MsMs*), and 15 plants were heterozygous (*Msms*) at the *Ms* locus. A total of 72 individual plants possessed S-cytoplasm and were homozygous recessive at the *Ms* locus (*msms*) and, therefore, could be used as female parents for the production of hybrid seed.

Only five plants with S-cytoplasm were scored as homozygous dominant at *Ms* (*MsMs*), and 32 plants were heterozygous (*Msms*). R-cytoplasm was identified in seven plants, of which three were homozygous recessive at the *Ms* locus (*msms*), two were homozygous dominant (*MsMs*), and two were heterozygous (*Msms*). T-cytoplasm was found in 30 plants, of which 29 were scored as homozygous recessive at the *Ms* locus (*msms*), and only one plant was heterozygous (*Msms*).

## 3. Discussion

To provide information on the distribution of CMS systems and to facilitate the monitoring of the cytoplasmic types of large breeding populations, we developed a two-step HRM marker system for the identification of N-, S-, R- and T-cytoplasms. The identification of the cytoplasm of a single plant takes from 4 to 8 years and is complicated by the segregation of the nuclear gene that restores fertility. The HRM marker system clearly distinguished the four types of cytoplasm. The availability of reliable markers for the selection of CMS and maintainer lines will accelerate the development of locally adapted hybrids [23]. The use of the HRM closed tube marker system with high sensitivity and high throughput greatly facilitates the work of breeders.

Analysis of 77 breeding lines (246 individual plants) showed the presence of individual plants that were heterozygous at the *Ms* locus within breeding lines. In total, 20.3% of individual plants were heterozygous at *Ms* (Table 3). One explanation could be that the dominant allele at *Ms* can show reduced penetrance, complicating efforts to purge dominant alleles from an inbred line or population [34]. Another reason for the high percentage of heterozygosity may be the non-use of molecular markers by breeders and selection by classical crossing and scoring segregation.

Eleven breeding lines were used for the development of the two-step HRM marker system. The fertility/sterility of the breeding lines were previously established by analysis of the anther morphology and microscopy of acetocarmine stained pollen grains [35]. The results of the HRM analysis of the cytoplasm types and *Ms* locus completely coincided with the data of previoulsy perfomed morphological and cytological analysis of the breeding lines (Table 4).

An interesting observation was that the breeding line ’Ivashka’ with R-cytoplasm was scored as homozogous dominant at *Ms* and is male fertile. This agrees with previous observations that the R-CMS is restored by the *Ms* locus, and this is consistent with the conclusion made by Havey and Kim [19]. We observed another scenario in the breeding line ’Odintsovets-37’ with T-cytoplasm, which was scored as homozogous recessive at *Ms* and fertile. However, if the gene/genes restoring the fertility of the T-cytoplasm were in the *Ms* locus, then the plant should be sterile with the *msms* nuclear genotype. Schweisguth [36] reported that male sterility of T–cytoplasm is conditioned by the interaction of T-cytoplasm with three male fertility restoration loci. Havey [37] showed by crossing the dominant restoration allele at *Ms* onto T-cytoplasmic male-sterile plants that T-cytoplasm is not restored by the *Ms* locus. Apparently, the three restoration loci that restore the fertility of the T-cytoplasm have yet to be discovered, and markers have yet to be developed.

Analysis of breeding lines from two germplasm collections showed that S-cytoplasm is the most common source of CMS used by Russian breeders. R-cytoplasm, a CMS developed from ’Rijnsburger’ onion and discovered by De Vries and Wietsma [38], appeared to be the rarest type of CMS among the Russian breeding lines, while R-cytoplasm is widely used to produce commercial hybrid seeds [19]. Of the 146 samples with sterile cytoplasm, 30 were with T-cytoplasm, which amounted to 20.5%. It is likely that the T-cytoplasm in the analyzed breeding lines originated from the Chalcedon variety created by breeders from the Pridnestrovian Agricultural Research Institute in the late 1980s and was then spread through the territory of the former Soviet Union. Most likely, the Chalcedon variety was created using the T-cytoplasm described by Berninger [17].

In this study, we developed a two-step HRM-based marker system for the identification of four onion cytoplasm. High-resolution DNA melting (HRM) has several advantages over conventional PCR, including an inexpensive closed tube format, i.e., no electrophoresis needed to read the PCR results. The system enables quick and easy distinguishing of the N-, S-, R- and T-types of onion cytoplasm. We expect the developed HRM markers will be widely used by breeders.

## 4. Materials and Methods

### 4.1. Plant Materials

77 breeding lines (246 individual plants) from the genetic collection of the Federal Scientific Vegetable Center and the genetic collection of the Timofeev Breeding Station were analyzed. In the third week of April, under favorable weather conditions and an air temperature of 14–16 °C, selected bulbs of 5–6 cm in diameter were planted in soil on 140 cm wide ridges in four rows (25 cm between rows and 15–20 cm between bulbs in a row). In the second week of May, when the growth of the leaves from the bulbs began, samples were taken from the young leaves. The leaves were stored in a freezer at –80 °C until DNA isolation.

### 4.2. Total DNA Extraction

Frozen samples were dried in a Labconco Drying Chamber and then pulverized in a TissueLyser II, Qiagene with stainless steel beads. The total genomic DNA was extracted using the cetyltrimethylammonium bromide (CTAB) method according to the protocol of [39].

### 4.3. Partial Sequencing of cox1 and orf725

Primers (EX-cox1-F, EX-cox1-R, EX-orf725-F and EX-orf725-R) were designed using Primer3Plus (https://www.primer3plus.com, accessed date 10 October 2022) to amplify gene fragments with a length 594 bp based on known *cox1* gene sequences extracted from the N-cytoplasm complete mitochondrial genome (GenBank: AP018390.1) and with a length 577 bp based on *orf725* gene sequences extracted from the CMS-S complete mitochondrial genome (GenBank: NC_030100.1). Since both genes are present in stoichiometric copies in the CMS mitochondrial genome, amplicons for each gene were obtained in separate PCR reactions.

The amplicons were Sanger sequenced (Evrogen, Moscow) using the designed primers. Poorly sequenced regions on the ends of amplicons were trimmed based on peaks in chromatograms. Only high-quality regions were used in the following analysis. Multiple alignments were created using Clustal Omega v1.2.4 [40] and visualized using Jalview v2.11.2.4 [41]. *cox1* and *orf725* gene sequences of *A. cepa* were extracted from normal (GenBank: AP018390.1) and CMS-S (GenBank: NC_030100.1) reference mtDNA sequences of *A. cepa*, respectively, based on annotation in the GenBank database.

### 4.4. AcM-HRM Analysis of Cytoplasm Type

High-resolution melting (HRM) PCR was performed using the primer combination of one common forward primer (AcM-HRM-F) and two specific reverse primers (AcM-HRM-R1 and AcM-HRM-R2). The AcM-HRM-F primer hybridizes in a conserved upstream region of both the *cox1* gene and the chimeric *orf725* gene derived from the *cox1* gene. The AcM-HRM-R1 primer anchors in the downstream region in the *cox1* gene and the AcM-HRM-R2 anchors in the downstream region of the *orf725* gene (Table 5).

A total volume of 20 μL of PCR mixture contained the following components: 2.5× RT-PCR reaction mix containing 2.5× PCR buffer B (KCl, TrisHCl, pH 8.8), 6.25 mM MgCl${}_{2}$, Syn Taq polymerase, dNTPs, Glycerol, Tween 20 and EVA Green (Syntol, Moscow, Russia), 0.5 μM of AcM-HRM F, 0.25 μM AcM-HRM R1 primer, 0.25 μM AcM-HRM R2 primer, 5 μL of DNA (0.05 μg) and sterile distilled water.

The PCR conditions were as follows: an initial denaturation step at 95 °C for 10 min and 45 cycles at 95 °C for 10 s, 60 °C for 5 s and 72 °C for 5 s. The PCR products were then heated to 95 °C with a ramp rate of 4.4 °C/s, cooled to 40 °C with a ramp rate of 2.2 °C/s and heated again to 65 °C with a ramp rate of 2.2 °C/s. By melting from 65 °C to 97 °C at a ramp rate of 0.07 °C/s, the melting curves were obtained.

### 4.5. T-HRM and atp9-HRM

A total volume of 20 μL contained the following components: 2.5× RT-PCR reaction mix containing 2.5× PCR buffer B (KCl, TrisHCl, pH 8.8), 6.25 mM MgCl${}_{2}$, Syn Taq polymerase, dNTPs, Glycerol, Tween 20 and EVA Green (Syntol, Moscow, Russia), 0.2 μM of dNTPs, 0.25 μM of T-HRM forward and reverse primers, 0.125 μM forward and reverse atp9-HRM primers, 5 μL of DNA (0.05 μg) and sterile distilled water.

PCR conditions were as follows, an initial denaturation step at 95 °C for 10 min and 45 cycles at 95 °C for 10 s, 61 °C for 10 s and 72 °C for 10 s. The PCR products were then heated to 95 °C with a ramp rate of 4.4 °C/s, cooled to 40 °C with a ramp rate of 2.2 °C/s and heated again to 65 °C with a ramp rate of 2.2 °C/s. By melting from 65 °C to 97 °C at a ramp rate of 0.07 °C/s, the melting curves were obtained.

High-resolution analysis of DNA melting curves was performed using a real-time PCR system on a Roche LightCycler 96 amplifier. Measurement of the kinetics of the dissociation of DNA molecules was performed using the EvaGreen intercalating dye (Lumiprobe, Moscow, Russia).

### 4.6. Conventional PCR

To check the HRM markers developed in this study, PCR was conducted with markers reported earlier by Engelke [25], Sato [24] and Kim et al. [26]. Primers that anchor in the upstream region to the *cob* gene were used [24] to distinguish between N-cytoplasm and CMS-S. Primers that span nearly the complete *orfA501* were used in combination with primers for the *cob* gene [24] as suggested from Engelke et al. [25] for the differentiation of all three types of cytoplasm. All primers are listed in Table 5.

PCR amplification was performed in a 25 μL reaction mixture containing 50 ng template, 2.5 μL 10× PCR buffer, 0.2 μL forward primer (10 μM), 0.5 μL reverse primer (10 μM), 0.5 μL dNTPs (10 mM each) and 0.5 U of Taq polymerase. The PCR products were visualized on 2% agarose gel after ethidium bromide staining.

### 4.7. HRM Analysis of the Ms Locus

Genotyping of the *Ms* locus was performed with HRM markers developed from full-length genomic DNA of *AcPMS1* sequences obtained from both homozygous dominant and recessive alleles as reported by Kim and Kim [30] (Table 5). The PCR mix and conditions were the same as described above for the AcM-HRM analysis of the cytoplasm type.

## Figures and Tables

**Figure 1 ijms-24-01605-f001:**
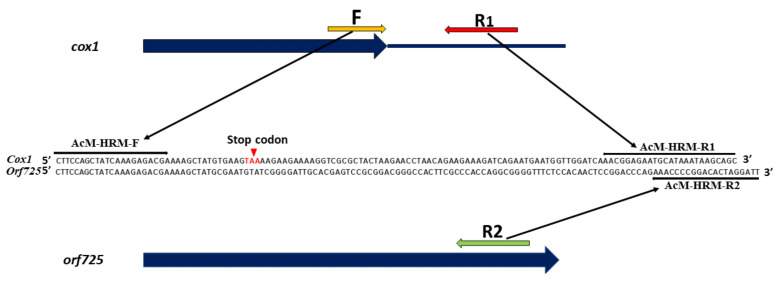
HRM primer annealing sites on *cox1* and *orf725* gene sequences. Primer binding sites are indicated by horizontal arrows. The red triangle indicates the stop codon of the *cox1* gene.

**Figure 2 ijms-24-01605-f002:**
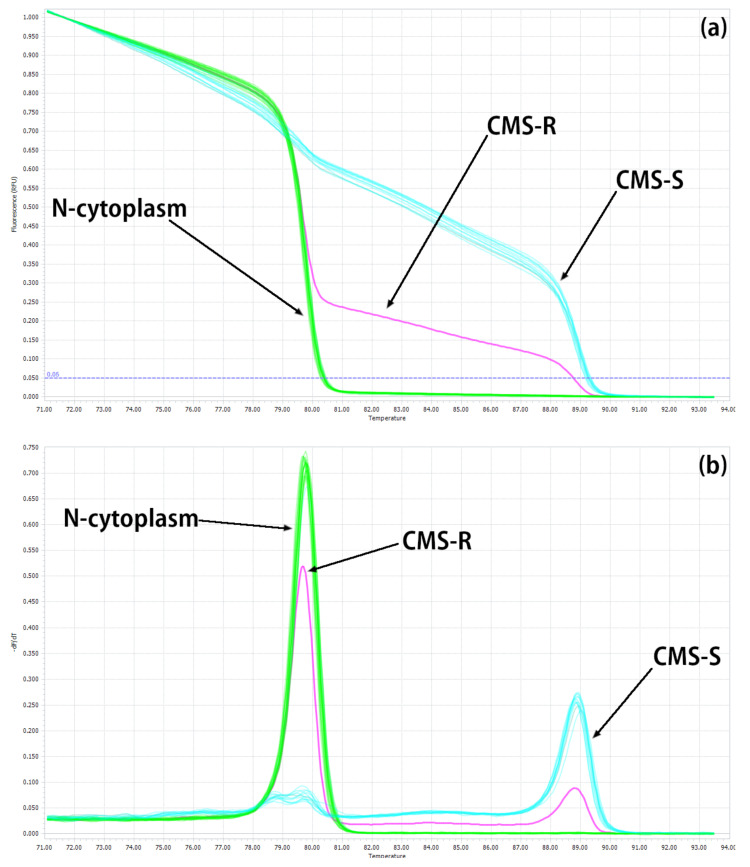
HRM analysis of cytoplasm type in onion using three AcM-HRM primers based on *cox1* and *orf725* mitochondrial genes: (**a**) normalized melting curve of amplicons and (**b**) normalized melting peaks of amplicons.

**Figure 3 ijms-24-01605-f003:**
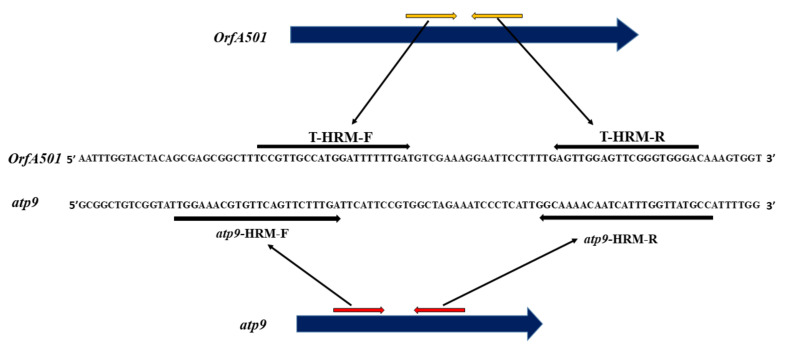
HRM primer annealing sites on *orfA501* and *atp9* gene sequences. Primer binding sites are indicated by horizontal arrows.

**Figure 4 ijms-24-01605-f004:**
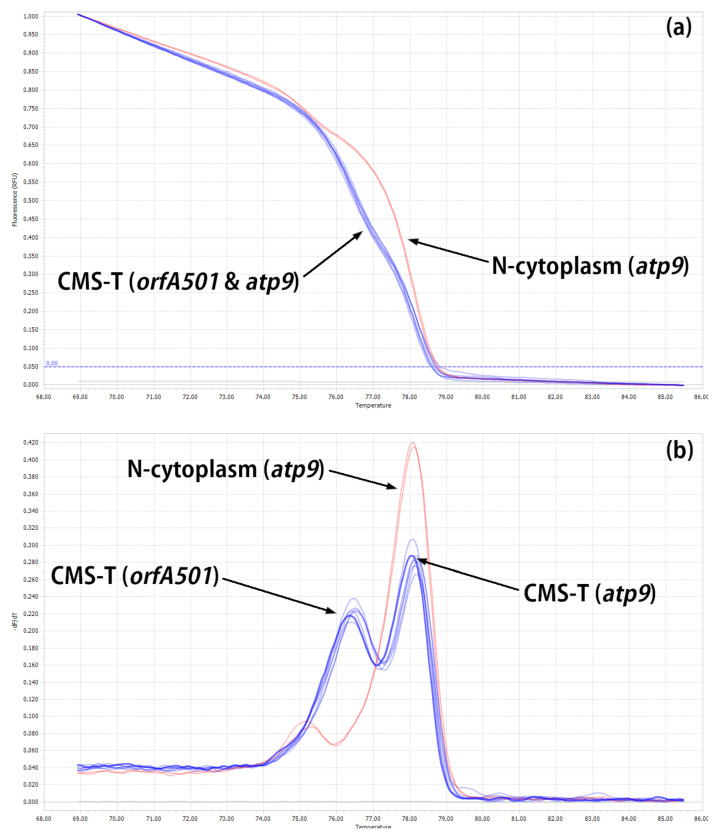
Differences between N- and T-cytoplasms detected using T-HRM primers based on *orfA501* and atp9-HRM primers based on the *atp9* gene (internal control): (**a**) normalized melting curve of amplicons and (**b**) normalized melting peaks of amplicons.

**Figure 5 ijms-24-01605-f005:**
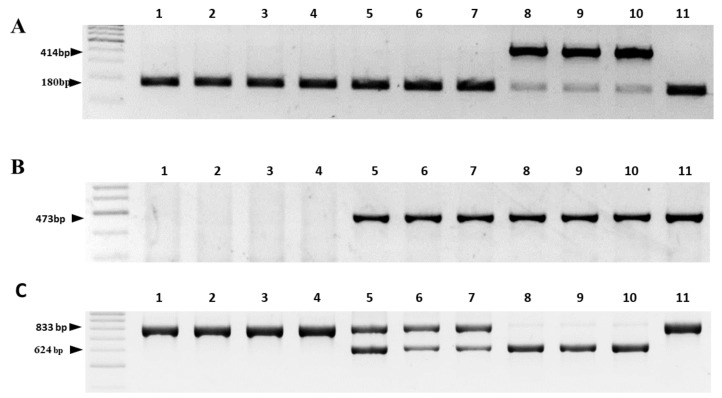
Identification of cytoplasm types in onion breeding lines using conventional PCR marker systems developed by Sato [24]—(**A**), Engelke et al. [25]—(**B**) and Kim et al. [26]—(**C**). Lanes: 1—Banko; 2—Sibirskiy; 3—Derek-8; 4—Odintsovets-28; 5—LFK; 6—CM Banko; 7—Ivashka; 8—Rawhide-17; 9—Sandra-276; 10—Derek-3; and 11—Odintsovets-37.

**Table 1 ijms-24-01605-t001:** Conventional PCR marker system in mitochondrial DNA distinguishing of onion cytoplasm according to the classification proposed by Heavy and Kim [19].

Reference	Marker on the Gene	Cytoplasm ^1^/Amplicon Size, bp			
		N	S	R	T
[24]	*cob*	180	414	180	180
[25]	*orfA501*	absent	473	473	473
[26]	*cox1*	833	absent	833	833
	*orf725*	absent	628	628	absent

1—Normal (N) male-fertility cytoplasm and male-sterile cytoplasm (S), (T) and (R).

**Table 2 ijms-24-01605-t002:** Comparative analysis of eleven breeding lines using conventional PCR and HRM analysis.

Breeding Line	Markers				Cytoplasm Type	
	*cob*	*orfA501*	*cox1*	*orf725*	PCR	AcM-HRM
Banko	180	absent	833	absent	N	N
Sibirskiy	180	absent	833	absent	N	N
Derek-8	180	absent	833	absent	N	N
Odintsovets-28	180	absent	833	absent	N	N
LFK	180	473	833	628	R	R
CM Banko	180	473	833	628	R	R
Ivashka	180	473	833	628	R	R
Rawhide-17	414	473	absent	628	S	S
Sandra-276	414	473	absent	628	S	S
Derek-3	414	473	absent	628	S	S
Odintsovets-37	180	473	833	absent	T	T

**Table 3 ijms-24-01605-t003:** Cytoplasm types and genotype of *Ms* locus in the onion breeding lines.

Cytoplasm Type	Genotype of *Ms* Locus			
	*MsMs*	*Msms*	*msms*	Total
N	12	15	73	100
S	5	32	72	109
R	2	2	3	7
T	0	1	29	30
Total	19	50	177	246
Percentage	7.7%	20.3%	72.0%	

**Table 4 ijms-24-01605-t004:** Comparison of pollen fertility, cytoplasm type and *Ms* locus data.

Breeding Line	Fertility/Sterility ^1^	Cytoplasm Type ^2^	*Ms* Locus *AcPMS1*
Banko	fertile	N	*msms*
Sibirskiy	fertile	N	*Msms*
Derek-8	fertile	N	*msms*
Odintsovets-28	fertile	N	*msms*
LFK	sterile	R	*msms*
CM Banko	sterile	R	*msms*
Ivashka	fertile	R	*MsMs*
Rawhide-17	sterile	S	*msms*
Sandra-276	sterile	S	*msms*
Derek-3	sterile	S	*msms*
Odintsovets-37	fertile	T	*msms*

^1^—Pollen fertility of analyzed lines was previously established by morphological analysis of anther development and microscopy of pollen [35]. ^2^—Normal (N) male-fertile cytoplasm and male-sterile cytoplasm (S), (T) and (R).

**Table 5 ijms-24-01605-t005:** PCR markers used for the determination of cytoplasm types and *Ms* locus in onion.

Primer Names	Primer Sequence (5’-3’)	The Marker Based on Genes	Reference
ACM-HRM-F	GCTATCAAAGAGACGAAAAGCT	*cox1* & *orf725*	This study
ACM-HRM-R1	GCTGCTTATTTATGCATTCTCCGT		
ACM-HRM-R2	AATCCTAGTGTCCGGGGTTT		
MK-F	CATAGGCGGGCTCACAGGAATA	*cox1* & *orf725*	[26]
MK-R1	AATCCTAGTGTCCGGGGTTTCT		
MK-R2	CAGCGAACTTTCATTCTTTCGC		
	ATGGCTCGCCTTGAAAGAGAGC	*orfA501*	[25]
	CCAAGCATTTGGCGCTGAC		
Forward primer	CTTTTCTATGGTGACAACTCCTCTT	*cob*	[24]
(S)-specific	GTCCAGTTCCTATAGAACCTATCACT		
(N)-specific	TCTAGATGTCGCATCAGTGGAATCC		
EX-cox1-F	TATCCAGATGCTTACGCCGG	*cox1* & *orf725*	This study
EX-cox1-R	ACTCGAACCTGCACTTCTGG		
EX-orf725-F	TTACGCCGGATGGAATGCTC		
EX-orf725-R	ACTGGGCGAATCACCACTTT		
T-HRM-F	TTTCCGTTGCCATGGATTTT	*orfA501*	This study
T-HRM-R	CCGAACTCCAACTCAAAAGG		
atp9-F	TGGAAACGTGTTCAGTTCTTTGA	*atp9*	
atp9-R	GCATAACCAAATGATTGTTTTGCCA		
Rf-HRM7-F	CCTATTCAATCCCTGGACATTT	*AcPMS1*	[30]
Rf-HRM7-R	GAGTTTGAAGGGCTATCTTTACTTG		

## Data Availability

Data available in a publicly accessible repository.

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
