# Peer review of "Two-Step Identification of N-, S-, R- and T-Cytoplasm Types in Onion Breeding Lines Using High-Resolution Melting (HRM)-Based Markers"

_ijms, 2023, doi:10.3390/ijms24021605_

Round 1

Reviewer 1 Report

In this paper the authors present the molecular methods for detection of the four types of CMS in Allium cepa.

The abstract is well written but the body of the article requires major revision.

The information from the introduction is not presented in logical sequence and does not provide enough details regarding the four CMS types in A. cepa.

There are errors in the legend and notations from the graphs. Example: Figure 2.a CMS-R in light blue is actually CMS-S.

Lines 157-166: it is not clear if the chimeric gene was actually constructed or it was in silico constructed.

Line 167:” A total of 130 individual plant DNA samples with N-cytoplasm as determined by the 167 first step” It is not clear what is the first step and from where are those 130 plants. In general it is not clear how many individual plants were analysed in this experiment.

In Discussion the data should be reorganized. For example: Line 217-221 the explanations are not clear and consistent. Line 225: it is not clear why it is interesting that a plant with R cytoplasm is fertile due to the presence of dominant Ms gene.

Not al the methods are described. For example, “1 - Pollen fertility was established by morphological analysis of anther development and microscopy of pollen grains using acetocarmine staining” is presented only as explanation for Table 3. We don’t understand why in 4.5 chapter (lane 293) RT-PCR reaction mix was used.

Author Response

We thank the Reviewer for a positive assessment of our research and useful comments and suggestions.

Please find our answers to comments (Your comments are indicated by dots)

  • The information from the introduction is not presented in logical sequence and does not provide enough details regarding the four CMS types in A. cepa.

When writing the introduction, the following logical sequence was built:

In the first paragraph (lines 15-22), we briefly introduce the reader to the object of study, its practical importance and the problem investigated.

In the second paragraph (lines 23-36) we give a brief background information about the CMS since the study is devoted to developing of markers that will allow distinguishing of CMS-types

In the third paragraph (lines 37-51) we briefly describe the plant mitochondrial DNA structure because CMS-associated genes are mitochondrial genes resulting from multiple recombination events.

In 4-5 paragraphs (lines 52-85) we describe in detail by whom and when three types of CMS (S, R- and T) were discovered, aberrant mitochondrial genes associated with specific types of CMS and markers available for distinguishing CMS types of cytoplasm.

.

  • There are errors in the legend and notations from the graphs. Example: Figure 2.a CMS-R in light blue is actually CMS-S.

Figure 2 was replaced with correct one according to comment.

  • Lines 157-166: it is not clear if the chimeric gene was actually constructed or it was in silico

Yes, the chimeric gene was constructed in silico. Changes have been made to the text.

  • Line 167: ”A total of 130 individual plant DNA samples with N-cytoplasm as determined by the 167 first step” It is not clear from where are those 130 plants …..how many individual plants were analysed 

We fully agree. This text is out of place and confuses the reader. We removed it.

The information about number of plant analyzed is presented in “Analysis of the cytoplasm types and Ms locus in breeding lines used by Russian breeders”

  • It is not clear what is the first step

Line 134 We added to the text:

So, in the first step, using three HRM-primers (AcM-HRM-F, AcM-HRM-R1, AcM-HRM-R2) were identified N-, S- and R-types of cytoplasm. However, these primers constructed on cox1 gene cannot distinguish between N (normal) cytoplasm and T –CMS, which sterility caused by rearrangement in other unknown mitochondrial gene.

  • . Not al the methods are described. For example, “1 - Pollen fertility was established by morphological analysis of anther development and microscopy of pollen grains using acetocarmine staining” is presented only as explanation for Table 3.

Reference for the method and changes to the text have been made

  • We don’t understand why in 4.5 chapter (lane 293) RT-PCR reaction mix was used.

HRM (High Resolution Melting) analysis is a type of post-PCR analysis which includes RT-PCR (Real Time-PCR) with following DNA melting, cooling and again slow melting.

The manuscript was checked by my colleague, a native English-speaking

Reviewer 2 Report

This work is interesting, however, there are some problems which can be addressed before final publication.

1.       Please provide the significance of the study in abstract (in one line)

2.       Follow the scientific rules when writing the species or genes names

3.       Please provide the purpose and importance at the end of the introduction

4.       Conclusion need further improvement

5.       There are various issues in the sentences like,

A). line No 15 Change following to "followed"

b). line No 21.     Write this Sentence like this "Knowledge about cytoplasm types plays an important role in onion breeding".

c). line 97. Has not clear meaning (Stoichiometries of cox1 and orf725 genes in plant cell define of N-, S- and R-cytoplasm types)

Author Response

We thank the Reviewer for a positive assessment of our research and useful comments and suggestions.

Please find our answers to comments (Your comments are indicated by dots)

  • Please provide the significance of the study in abstract (in one line)

We added text at the end of the abstract

  • Follow the scientific rules when writing the species or genes names

All genes' and species' names were checked and corrected if necessary.

  1. line No 15 Change following to "followed"

Corrections have been made to the text

  1. line No 21. Write this Sentence like this "Knowledge about cytoplasm types plays an important role in onion breeding".

Corrections have been made to the text

  • Line 97 line 97. Has not clear meaning (Stoichiometries of cox1 and orf725genes in plant cell define of N-, S- and R-cytoplasm types)

mtDNA is composed of a mixture of circular, linear and branched molecules of different sizes that are capable of recombination. The ratio of the number of normal genes (N-cytoplasm) and their aberrant versions (CMS) determines the type of cytoplasm.

  • Conclusion need further improvement

We added text at the end of the Discussion

The manuscript was checked by my colleague, a native English-speaking

Round 2

Reviewer 1 Report

RT-PCR means Reverse Transcription PCR not Real Time PCR.